# Sequestration of Inflammation in Parkinson’s Disease via Stem Cell Therapy

**DOI:** 10.3390/ijms231710138

**Published:** 2022-09-04

**Authors:** Jonah Gordon, Gavin Lockard, Molly Monsour, Adam Alayli, Hassan Choudhary, Cesario V. Borlongan

**Affiliations:** 1Morsani College of Medicine, University of South Florida, Tampa, FL 33602, USA; 2Center of Excellence for Aging and Brain Repair, Department of Neurosurgery and Brain Repair, College of Medicine, University of South Florida, Tampa, FL 33612, USA

**Keywords:** Parkinson’s disease, stem cell, neuroinflammation

## Abstract

Parkinson’s disease is the second most common neurodegenerative disease. Insidious and progressive, this disorder is secondary to the gradual loss of dopaminergic signaling and worsening neuroinflammation, affecting patients’ motor capabilities. Gold standard treatment includes exogenous dopamine therapy in the form of levodopa–carbidopa, or surgical intervention with a deep brain stimulator to the subcortical basal ganglia. Unfortunately, these therapies may ironically exacerbate the already pro-inflammatory environment. An alternative approach may involve cell-based therapies. Cell-based therapies, whether endogenous or exogenous, often have anti-inflammatory properties. Alternative strategies, such as exercise and diet modifications, also appear to play a significant role in facilitating endogenous and exogenous stem cells to induce an anti-inflammatory response, and thus are of unique interest to neuroinflammatory conditions including Parkinson’s disease. Treating patients with current gold standard therapeutics and adding adjuvant stem cell therapy, alongside the aforementioned lifestyle modifications, may ideally sequester inflammation and thus halt neurodegeneration.

## 1. Introduction

Parkinson’s disease (PD) is one of the most common neurodegenerative disorders, with an incidence rate of 61.21 and 37.55 per 100,000 people in males and females over 40, respectively [1]. The determinants of this disease continue to be investigated; however, its pathophysiology appears to be multifactorial, with various genetic components [2,3,4] and environmental factors, such as chemical exposure, increasing one’s risk of acquiring the disease [5]. The disease is classically characterized by the death of dopaminergic (DA) neurons in the basal ganglia, but the neurodegeneration extends beyond this structure, causing symptoms such as bradykinesia, resting tremors, and rigidity [6,7]. The aggregation of α-synuclein in neurons during the development of PD has been implicated as a critical cell death mechanism, although the exact role such aggregates contribute to its pathogenesis remains to be fully elucidated [8,9,10,11]. A crosstalk may exist between the activation of microglia and the overaccumulation or improper folding of α-synuclein. The neurodegenerative cascade may initiate with the migration of microglia to the site of aggregation, where they phagocytose the α-synuclein and induce an inflammatory immune response with downstream effects [12,13].

Inflammatory processes play a major role in the development of PD pathology and its symptoms. Indeed, a high concentration of glutathione peroxidase in glial cells surrounding DA neurons suggests a protective response to oxidative stress [4]. In contrast, higher levels of microglia in older mice are associated with significant neurodegeneration in the substantia nigra [14]. Microglia are key mediators in neuroinflammation, so this finding offered new insight into the role of inflammation in PD [15]. Activated microglia release a flood of cytokines and pro-inflammatory molecules such as IL-1, TNF-α, nitric oxide (NO), prostaglandin E_2_, and superoxide, all of which can have wide-ranging and damaging downstream effects [16]. Upon the deeper investigation of these cytokines, scientists note that inhibiting inducible nitric oxide synthase, the enzyme responsible for NO production, in rat models of PD induces a significant reduction in neurotoxicity, as shown by the increased integrity of the nigrostriatal pathway and decreased microglial count when compared to control rats [17]. Similarly, NADPH-oxidase participates in the development of neurotoxicity in rat PD models [18]. Aside from microglia, T cells circumvent the blood–brain barrier, infiltrating the brains of PD patients along with splenocytes [19,20]. Indeed, the increased propagation of microglia closely accompanies infiltrating T cells, suggesting that microglia act as the antigen-presenting cells to the T cells after they have crossed the blood–brain barrier [21].

A genetic basis may also mediate the aberrant inflammation in the development of PD. Mice with leucine-rich repeat kinase 2 (LRRK2) knock-down display an inhibited microglial inflammatory response when compared to control mice [22]. On the other hand, mice with DJ-1 knock-down astrocytes exhibit significantly more NO production and demonstrate apoptotic behavior, indicating that the gene plays an important role in mediating the inflammatory response [23]. Other genes associated with inflammation and neurotoxicity include PINK1, Parkin, and others not mentioned in this review [24,25,26]. Any attempt to regulate the inflammatory mechanisms in PD should recognize the complexity and multifactorial nature of neuroinflammation.

The aim of this review is to explore the most recent research on the topic of neuroinflammation, which seeks to understand the role of stem cell therapy as a potential therapeutic mediator of the inflammatory damage caused during the natural progression of PD. As later sections will discuss, current therapies can often exacerbate neuroinflammation, and thus finding a treatment that can sequester inflammation and halt the progression of PD would be a major development in the management of the disease.

## 2. Pathophysiology and Therapeutics for Parkinson’s Disease

In the basal ganglia, there are two motor pathways: the direct pathway and the indirect pathway. The direct pathway is excitatory for movement, and involves the striatum inhibiting the globus pallidus internus (GPi) and thus attenuating the inhibitory effect of the GPi on the thalamus. The thalamus is excitatory to the motor cortex, initiating movement. Contrarily, the indirect pathway is inhibitory for movement. The striatum inhibits the globus pallidus externus (GPe), thus attenuating the inhibitory effect of the GPe on the subthalamic nucleus (STN). The STN then excites the GPi, which inhibits the thalamus. The excitatory thalamic signaling to the motor cortex is lost, resulting in the inhibition of movement.

Both the indirect and direct pathways can be impacted by dopamine neurotransmission. Dopamine is an endogenous neurotransmitter, critical for the movement mediated by the basal ganglia. Dopamine is a catecholamine and is derived from L-DOPA via the enzyme DOPA decarboxylase, which itself is derived from L-tyrosine via the enzyme tyrosine hydroxylase. The substantia nigra pars compacta, located in the midbrain, releases dopamine onto the striatum via the nigrostriatal pathway. Dopamine binds to the D1 and D2 receptors in the striatum; subsequent D1 signaling stimulates the direct (excitatory) pathway, and D2 signaling inhibits the indirect (inhibitory) pathway. Ultimately, this signaling encourages movement initiation.

As noted above, PD presents with the degeneration of DA neurons in the pars compacta, which severely affects nigrostriatal signaling. To ameliorate the motor symptoms of PD, such as tremor, rigidity, and bradykinesia, exogenous therapy is utilized. Unfortunately, dopamine cannot cross the blood–brain barrier. Thus, a dopamine precursor, L-DOPA, is administered as the prodrug levodopa, which is converted in the central nervous system to dopamine via DOPA decarboxylase. Of note, levodopa is co-administered with carbidopa, which inhibits DOPA decarboxylase activity in the peripheral tissues, with the goal of increasing levodopa’s bioavailability in the brain. Another therapeutic option is deep brain stimulation (DBS). DBS was originally used to localize the target of a resection, but now subcortical neurostimulation is utilized as the first-line surgical treatment of movement disorders [27,28]. DBS stimulation, commonly in the STN or GPi, can modulate signaling within the basal ganglia, thalamus, and motor cortex to improve the motor dysregulation seen in PD.

Inflammation in the setting of PD contributes to the neurodegenerative process and has both acute and chronic phases. As PD worsens, reactive microglia increasingly localize in the substantia nigra, putamen, hippocampus, and cortex [29,30,31,32,33]. As chronic neuroinflammation progresses, microglia lose their reparatory M2 phenotype, and preferentially differentiate to their inflammatory M1 phenotype [34,35]. IL-6 is a proinflammatory cytokine local to the CNS, released by neurons, astrocytes, endothelial cells, and microglia [36]. IL-6 concentrations are positively correlated with motor scores in PD patients, as measured using the Movement Disorders Society-Unified Parkinson’s Disease Rating Scale (MDS-UPDRS) [37]. PD patients with fatigue have significantly increased serum levels of IL-6 compared to PD patients without fatigue, as measured with the Parkinson’s Disease Fatigue Scale. Fatigued patients were reported to have worsened signs and symptoms of PD and increased functional dependence [38]. PD patients with higher IL-6 concentrations at baseline were seen to have worsened depression scores two years later [39]. There is a negative correlation between IL-6 levels and scores on the Activities of Daily Living scale, indicating that patients with worsened PD have greater IL-6 concentrations [40]. Interestingly, the literature indicates microglia and inflammatory cytokines such as IL-6 are involved in both neurodegeneration (typically in chronic settings) and neuronal survival (typically in acute settings) [33,35,41].

Unfortunately, L-DOPA and DBS therapy may worsen neuroinflammation. It is almost ironic that chronic L-DOPA therapy may precede L-DOPA induced dyskinesia (LID) [42]. After L-DOPA administration, there is excessive dopamine that saturates vesicles, resulting in a plethora of free cytosolic dopamine that is degraded by monoamine oxidase, leading to an accumulation of cytotoxic reactive oxygen species [43,44]. L-DOPA in the setting of PD leads to aberrant D1 signaling in the striatum, with the subsequent hyperactivation of cAMP and PKA [45]. In LID rat models, glucocorticoid treatment reduced abnormal involuntary movements [46], demonstrating L-DOPA’s role in the neuroinflammation seen to worsen Parkinsonism dyskinesia. In the same model, glucocorticoids also normalized the increased expression of proinflammatory cytokines, such as IL-1β, in the striatum. Additionally, the direct administration of IL-1 receptor antagonists to the striatum reduced abnormal involuntary movements [46]. Hepcidin, an acute phase reactant synthesized in the liver, is upregulated by IL-6 in settings of inflammation and is responsible for decreasing iron absorption through the destruction of ferroportin and the reduction in iron release from macrophages. Pro-hepcidin serum concentrations were significantly higher in PD patients treated with DBS and pharmacological therapy than in PD patients treated with pharmacological therapy only and in control patients [47,48].

Figure 1 demonstrates the aforementioned contribution of levodopa–carbidopa and deep brain stimulation to neuroinflammation.

## 3. Rewiring the Inflammatory Responses

### 3.1. Anti-Inflammatory Based Therapies

While most existing therapies for PD focus on restoring DA neuron loss or using DBS to restore basal ganglia pathways, evolving evidence for PD as an inflammatory disease prompts the use of anti-inflammatory therapies to ameliorate PD progression [49,50]. Several studies in clinical and pre-clinical populations have demonstrated a reduced PD risk and attenuated striatal DA neuron loss using anti-inflammatory drugs, such as ibuprofen, acetylsalicylic acid, and NSAIDs [51]. Focusing on immune cell proliferation, other groups have designed dendritic cell (DC) vaccines to modulate the typically inflammatory immune response to α-synuclein. In PD, DCs phagocytose toxic α-synuclein buildup to present to lymphocytes in the nearby cervical lymph nodes. T cells respond to the toxic α-synuclein and induce a pro-inflammatory environment [52,53] composed of M1 microglial and A1 astrocytic differentiation and inflammatory cytokines [54,55]. Using primed DCs, differentiation can be swayed towards anti-inflammatory M2 microglia, A2 astrocytic, and Th2 CD4+ T-cell phenotypes [54]. In mice, locomotor function is improved after sensitizing DCs to α-synuclein [56]. Decreased blood DCs are correlated with worse motor impairment, further linking the potential for DC vaccines in modulating PD prognosis [57]. Antibodies also demonstrate similarly successful immunomodulation in animal models of PD [58,59,60]. The administration of anti- α-synuclein antibodies prompts microglial clearance in mouse models of PD, reduces neuronal and glial protein accumulation, and translates to improved functionality [61]. In clinical studies, the administration of α-synuclein monoclonal antibodies, BIIB054 and PRX002, has also shown encouraging results [62,63,64]. Thus, anti-inflammatory pharmaceuticals may be important therapies to consider in treating PD, either alone or in combination with other PD therapeutics.

### 3.2. Cell-Based Therapies

Another promising route for inducing anti-inflammatory responses in PD may use stem cells. Many of the therapeutic effects of stem cells arise from their inherent secreted anti-inflammatory signals among other cell survival molecules, deemed the bystander effects [65]. Mesenchymal stem cells (MSCs), for example, demonstrate anti-inflammatory, anti-apoptotic, and regenerative capabilities [66,67]. ESC–MSCs and iPSC–MSCs can inhibit lymphocyte and natural killer cell proliferation, also showing promise for immunoregulatory cell-based therapies [68,69,70,71]. As noted above, exosomes show promise for PD therapies [66]. MSC-derived exosomes have anti-inflammatory, antioxidant, and neurotrophic properties, ultimately leading to functional recovery in animal models of PD [72,73]. An appealing feature of stem cell treatments includes their ability to be modified prior to implantation, for instance, to promote anti-inflammatory actions. A pathway of interest includes interferon-γ signaling via the aforementioned gene, LRRK2. LRRK2 G2019S is the most common genetic mutation associated with inherited PD. In PD patient-derived iPSCs possessing LRRK2 G2019S, IFN-γ signaling increases immunomodulated neuronal damage [74]. MSCs are initially activated by IFN-γ but then exert anti-inflammatory properties [75]. Thus, MSCs may effectively modulate this pathological signaling.

Furthermore, recognizing the underlying DA neuron loss in PD, it is logical to develop treatments that may reestablish these neuronal populations and connections. Cell-based therapies, however, show mixed results in PD, suggesting that additional anti-inflammatory enhancements of the cells may improve their viability. The use of grafts to restore DA neurons shows mixed results, with some reports demonstrating survival, innervation, and improved functionality, and others showing no clinical benefit [76,77,78,79,80,81,82,83,84]. The studies demonstrating the therapeutic value of stem cells for PD, however, prompt further investigation [85,86]. For instance, exosomes from human umbilical cord mesenchymal stem cells (hucMSCs) reduce apoptosis and encourage proliferation in an in vitro PD cell culture. In rats, hucMSC treatment shows a similar prevention of DA neuron loss and recovered DA levels [66]. To modulate the altered DA signaling, GABAergic cells from the medial ganglionic eminence (MGE) can also be targeted. MGE cells show potential to migrate and differentiate into GABAergic neurons, restoring function to mouse models treated with 6-hydroxydopamine (6-OHDA), a selective neurotoxin of DA neurons to mimic PD [87]. Human embryonic stem cells (hESCs) in rat models of PD can project long distances in the rodent brain to mitigate symptoms, offering hopeful clinical translation [88]. Even closer to the goal of clinical application, human parthenogenetic embryonic stem cell (hPESC)-derived DA neurons improve mobility in monkey models of PD [89]. Additionally, in monkey 1-methyl-4-phenyl-1,2,3,6-tetrahydropyridine (MPTP) models recreating PD, human-induced pluripotent stem cells (iPSCs) improve movement and demonstrate ample neurite formation. Whether grafted from PD or control donors, iPSCs survive, make connections, and lead to functional recovery in monkeys [90]. There is still room for improvement in stem cell PD treatments, as full recovery is not typically achieved in animal models [91,92,93].

A comprehensive list of pre-clinical anti-inflammatory stem cell studies for PD, along with the cell types, dosages, and findings, is outlined in Table 1. The studies were completed in rodent models in vivo (6-OHDA or MPTP technique) or in vitro (A53T α-synuclein overexpressing cells).

### 3.3. A Potential Synergistic Niche

While already anti-inflammatory by nature, stem cell treatments may be further enhanced in clinical effectivity and viability by modulating these cells to express additional anti-inflammation properties. The potential for MSCs to impact cell signaling can also be applied to the Wnt/β-catenin pathway. The Wnt/β-catenin signaling pathway is involved in promoting DA neurorepair [101]. In patients with a LRRK2 mutation, the neurodegenerative phenotype may arise from altered Wnt signaling. This alteration increases the inhibitory phosphorylation of β-catenin using GSK-3β [102]. Normally, Wnt signaling functions to upregulate β-catenin, and to promote the transcription of genes associated with cellular survival, proliferation, and differentiation [103]. Wnt signaling can be stimulated in PD treatment models using chemical Wnt activators to induce the differentiation of iPSCs to midbrain dopaminergic (mDA) neurons [86,104]. Further enhancing the stimulation of iPSCs into mDA neurons can be carried out through the co-activation of both canonical and non-canonical Wnt pathways. The overexpression of Wnt3a or Wnt5a has been shown to significantly promote ESC differentiation to DA neurons. Compared to the activation of Wnt signaling with only FGF2/FGF8/Shh, adding Wnt5a overexpression enhanced DA neuron production 4-fold [105]. Wnt3a is a component of the canonical route of mDA neurogenesis, and when overexpressed, prevents differentiation from reaching its full potential. However, when Wnt3a is administered first to an undifferentiated cell line to promote proliferation, followed by a “wash-out” phase and then exposure to Wnt5a to promote cell cycle exit and Nurr+ differentiation, a large rise in Tyrosine Hydroxylase + (TH+) DA neurons is seen [105,106]. The neuroregenerative signaling associated with Wnt/β-catenin is further established in MPTP-induced PD mice models. Transplanting neural stem cells (NSCs) to host substantia nigra pars compacta (SNpc) results in a differentiation of 30% of NSCs to the astrocyte phenotype, allowing the host microenvironment of SNpc to enhance DA neuronal plasticity [97,107,108]. Furthermore, it should be noted that through the antagonism of Wnt/β-catenin signaling in MPTP mice, the therapeutic effects of the transplanted NSCs were abolished [107,108]. In line with previously discussed benefits of anti-inflammatory modulators [65], NSC-mediated Wnt/β-catenin signaling downregulates microglia and reduces inflammatory mRNA species, including Tnf and Tnfrsf1a, Il1, Nos2, Nfkb, and phagocyte oxidase Cybb (gp91phox) [97]. The potential for unique therapeutic effects from NSC grafts may be realized considering the growing findings of the Wnt/β-catenin pathway in PD [109,110]. The use of cell therapies to amplify anti-inflammatory effects is a promising direction for future PD therapies given the immense neuroinflammation and dopaminergic cell death contributing to the disease’s pathophysiology.

### 3.4. Neurotrophic Factors and Stem Cell Therapy

Neurotrophic factors may demonstrate protective properties in the setting of stem cell therapy and neurodegenerative disease. Of particular interest, glial cell line-derived neurotrophic factor (GDNF) nurtures the survival of dopaminergic neurons commonly affected in PD [111]. GDNF signaling is complicated, but importantly, it binds to GDNF family receptor 1α, which subsequently binds to RET, promoting neuronal survival and plasticity [112]. Murine and primate dopaminergic neurons were spared from 6-OHDA- and MPTP-induced cell death upon treatment with GDNF [113,114,115]. Due to the success in animal models, seven human clinical trials have been completed; GDNF was administered, often via putamen infusion, and UPDRS III scores were measured [116,117,118,119,120,121,122]. The results were unfortunately ambivalent; some patients had dramatic improvements in motor function, but in the three studies with placebo control groups, there were no significant differences compared to the treatment group. Neurotrophic factors are a worthy consideration in the treatment of PD through interactions with stem cells.

## 4. Lifestyle Factors Facilitating the Stem Cell Response

### 4.1. Exercise and Physical Therapy

Exercise and physical therapy are critical strategies known to facilitate endogenous and exogenous stem cells in inducing an anti-inflammatory response, which thus offer potential benefits alongside concomitant stem cell therapy for PD [123,124,125,126,127,128].

Physical therapy and exercise training are significantly implicated in anti-inflammatory activity independent of age [129,130] or overall health status [131]. Physical activity shows benefits in many chronic conditions, such as cardiovascular disease [132,133], cancer [134,135], and neuroinflammatory diseases [136]. Even exercise surrogates, such as whole-body vibration training, are shown to attenuate inflammation and mobilize stem cell populations associated with vascular health [137]. Further, a recent systematic review and meta-analysis highlights the anti-inflammatory role of exercise when controlling for variables such as type, intensity, and volume of exercise, suggesting the mechanism is related to an induction of antioxidant indicators [138]. In mouse models, exercise is implicated in stem cell driven anti-inflammatory effects through changes to the epigenome and transcriptome of hematopoietic stem and progenitor cells, diminishing the proliferative yield of inflammatory immune cells [139]. Importantly, this finding is subsequently noted in adult populations independent of BMI [140]. The effects on neuroinflammation are mediated through the downregulation of microglia via the direct inhibition or suppression of pro-inflammatory cytokines released from a number of cell types [141]. Exercise in both animal models and PD patients has shown promise for lessening the neuroinflammatory processes associated with PD. A rat model receiving unilateral striatal injections of 6-OHDA to induce PD demonstrates that treadmill exercise downregulates the activation of microglia, astrocytes, and oxidative species, and indicates that regular exercise may preserve dopamine levels in PD [142]. This mechanism of reduced neuroinflammation is further elucidated in MPTP-induced mice, in which treadmill exercise downregulates the expression of Toll-like receptor 2 and its downstream molecules, resulting in reduced inflammatory markers, such as TNF-α, NF-κB, IL-1β, and NADPH oxidase, while enhancing plasma dopamine levels and dopamine transporter expression [143]. Moderate exercise also demonstrates reduced levels of TNF-α in patients with PD [144]. Treadmill activity in 6-OHDA rats is also seen to enhance the expression of BDNF, SERCA II, superoxide dismutase, and catalase, all of which decrease neuroinflammation and promote an antioxidant or regenerative response [145].

In addition to neuroinflammatory modulation, exercise has been shown to enhance the viability of endogenous NSCs. Physical exercise is shown in a number of rodent models to trigger the proliferation of endogenous NSCs [146,147,148]. Moderate exercise in mouse models with inflammation-induced hippocampal suppression specifically restores neurogenesis in the dentate area of the hippocampus [149]. Exercise also induces the release of a number of the neurotrophic factors described above, with GDNF concentrations particularly increasing in 6-OHDA rats, MPTP mice, and MPTP monkeys to promote endogenous cell survival for vulnerable dopaminergic neurons [150]. Further, MPTP mouse models placed on an exercise regimen show improved neurogenesis in the subventricular zone, subgranular zone, substantia nigra, and striatum [151]. These effects have been replicated, to a moderate extent, in studies on human PD patients. In these patients, intensive exercise elevates serum BDNF by 16%, enhancing the overall proliferative capacity of endogenous NSCs [152].

Exogenous stem cells also show significant benefits pertaining to exercise therapy. For instance, exercise in mouse models undergoing bone marrow transplantation results in enhanced stem cell survival and recipient blood cell reconstitution [153]. Additionally, physical therapy is proven to be a key aspect linked to inflammation and survival rates in cancer patients undergoing hematopoietic stem cell transplantation [154]. Transplanted exogenous GABAergic neural progenitor cells result in neuropathic pain reduction when coupled with intensive locomotor training [155]. When taken together, exercise therapy is, therefore, a promising field for future research into PD with potential to sequester inflammation-associated cell death and promotes both an endogenous and exogenous stem cell response.

### 4.2. Diet and the Gastrointestinal Microbiome

Dietary consumption has recently taken a prominent role in research into neuroinflammation and PD, in which food intake, supplementation, and the gut microbiome may have significant therapeutic potential for decreasing inflammation and enhancing stem cell survival and proliferation.

Amassing evidence indicates the role of diet in suppressing neuroinflammation. The ketogenic diet, consisting of high fats, low proteins, and low carbohydrates, has garnered interest for potential roles in neuroprotection and the inflammatory response. In MPTP-induced mice, the ketogenic diet is shown to be both neuroprotective and anti-inflammatory, alleviating motor dysfunction and inhibiting microglial activation, thereby reducing proinflammatory cytokines in the substantia nigra [156]. This underlying effect is likely attributed, in part, to the elevation of the ketone β-hydroxybutyrate, which has been shown to be neuroprotective for dopaminergic models and involved in the downregulation of microglial-induced neuroinflammation [157], having been documented as blocking the NLRP3-inflammasome [158]. Such ketone bodies have also been shown to reduce oxygen-free radicals of the mitochondrial respiration chain in DA neurons, further sequestering PD-associated inflammation [159,160]. Pilot studies in PD patients demonstrate an early confirmation of the role of ketosis in enhancing cognitive performance, though have failed to impact motor function [161]. A number of other dietary patterns and supplementations implicate a role for diet in the neuroinflammation and protection of neural stem cells. For instance, omega-3 fatty acids and vitamin E co-supplementation in PD patients show reduced inflammation (as measured by C-Reactive Protein) and enhanced total antioxidant response, thereby promoting neuron survival [162]. Polyphenols found in green tea have been studied within PD models, with potential mechanisms that include preventing α-synuclein aggregation (restoring the normal differentiation and apoptotic suppression of endogenous precursor cells associated with the protein) [163,164], increasing tyrosine hydroxylase and dopamine levels [165], and sequestering iron (known to trigger oxidative stress, α-synuclein aggregation, inflammation, and neurodegeneration) [166,167]. Resveratrol, a stilbene produced by various fruits and vegetables, is shown to mitigate neuroinflammation in the substantia nigra of PD-induced rats through the downregulation of TNF-α and COX-2 [168].

Additionally, dietary patterns have been implicated in the endogenous stem cell response. For example, intermittent fasting may possess utility in the management of PD, in which mouse models subjected to alternate day fasting prove to be more resistant to the MPTP induction of PD, potentially indicating a neuroprotective effect on existing DA neurons or the enhanced proliferation of precursor cells [169]. Generally, dietary restriction appears to stimulate endogenous stem cell neurogenesis and enhance synaptic plasticity [170], which could represent an important mechanism for resisting injury associated with PD. It is important to note that diets high in antioxidants may also benefit exogenous stem cell survival. For instance, the survival of grafted fetal ventral mesencephalic tissue is increased when a Sprague Dawley rat model of PD is given a diet of 2% blueberries, a fruit high in antioxidants. Therefore, these adjunctive diet regimens can ultimately be applied to human grafting as a PD combination treatment to promote overall stem cell viability [171].

However, diet can also have negative implications for neuroinflammation and PD. Wistar rat models on a high-fat diet for 25 weeks to model obesity demonstrate decreased tyrosine hydroxylase and increased microglial activity, TNF-α, oxidative stress and astrogliosis [172]. Similarly, a mouse model on a similar diet yields a loss of DA neurons in the substantia nigra [173]. Nevertheless, given the easily implicated and accessible benefits of certain dietary adjustments, this may be a promising initiative to lessen neuroinflammation and the symptoms of PD.

In recent years, the gastrointestinal microbiome has been found to be increasingly connected with inflammatory processes and the central nervous system. The neurodegeneration association with PD is associated with a shift in the gut microbiome towards a pro-inflammatory environment with bacteria that produce endotoxins (such as LPS) and participate in methanogenesis [174,175]. While the precise mechanism of this interaction is not fully understood, it is known that there exists bidirectional communication between the enteric and central nervous systems, known as the gut–brain axis, which allows for brain lesions to arise from the gut microbiome [176]. As further evidence of this peripheral inflammation infiltrating the brain, recent experimental mouse models show that glial activation is associated with the infiltration of peripheral adaptive immune cells [177], with Th17 lymphocytes invading the substantia nigra and enhancing DA neuron death [178]. Thus, treatment directed towards improving the microbiome, or intake that disrupts it, could have profound implications for PD. For instance, antibiotic and antifungal exposure is shown to increase the risk of PD through microbiome dysbiosis [179]. Meanwhile, a recent case report describes a fecal microbiota transplantation to relieve intractable constipation, an important symptom of PD, which resulted not only in unobstructed defecation but also a nearly two-month reprieve from tremors in the lower extremities [180]. In MPTP-induced mice, such a procedure exhibits neuroprotection through the inhibition of glial cells and neural inflammation [181]. Additionally, smoking and coffee consumption contribute to a reduced risk of PD development over the life course through their impact on the gut microbiome [182]. Probiotics have become important to PD research to mitigate peripheral inflammation, with studies in patients resulting in improved bowel symptoms and improved disease states according to the MDS-UPDRS [183,184,185,186,187]. Overall, diet represents an important approach in mediating inflammation to promote neural stem cell survival and proliferation, while simultaneously opening newfound research avenues related to modulating gut dysbiosis toward the sequestration of peripheral, as well as central, inflammation.

## 5. Conclusions and Future Directions

This review summarizes current and experimental treatments for PD, with a focus on stem cell therapies that address the neurotoxicity and inflammation that is prevalent throughout the disease’s progression. The neuroinflammation seen in the disease is the product of the crosstalk between local and peripheral inflammatory pathways, which play a critical and interwoven role in its pathophysiology. Current gold standard treatments, such as levodopa–carbidopa and DBS, can exacerbate inflammation, so patients may benefit from anti-inflammatory cell-based treatments, such as the transplantation of human stem cells or stem-cell-derived DA neurons. Aside from their anti-inflammatory properties, these treatments benefit from their inherent versatility, in some cases allowing them to both downregulate inflammation and simultaneously promote regeneration. Combining existing therapies with emerging stem cell treatments and lifestyle modifications that promote the stem cell-meditated sequestration of neuroinflammation appears to be a promising new direction that warrants further investigation.

## Figures and Tables

**Figure 1 ijms-23-10138-f001:**
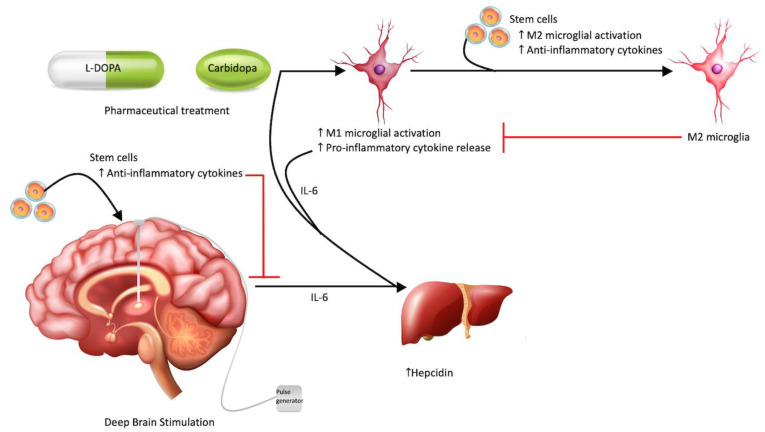
Therapeutic approaches to Parkinson’s disease. Typical treatment modalities for Parkinson’s disease, such as DBS and pharmaceuticals (L-DOPA and Carbidopa), may amplify the pro-inflammatory environment of PD by preferentially stimulating M1 microglial differentiation and inflammatory cytokine release. These cytokines can also increase hepcidin, an acute phase protein, from the liver. Stem cells may act via the bystander effect to minimize this neuroinflammation by shifting microglial differentiation to M2 and increasing anti-inflammatory cytokines.

**Table 1 ijms-23-10138-t001:** Pre-clinical anti-inflammatory stem cell studies for Parkinson’s disease.

Citation	Model	Cell type	Dosage	Findings
Haney et al. (2013) [94]	6-OHDA mice	Modified macrophages with antioxidant plasmid	5 × 10^6^ cells/mouse in 100 µL PBS	Macrophages released exosomes containing antioxidant genetic material prompting neurons to increase protein synthesis. Mice had improved motor function.
Ugen et al. (2015) [56]	A53T α-synuclein overexpressing cells and mice	Bone marrow-derived dendritic cells	10^6^ cells	α-synuclein sensitized DCs induced α-synuclein antibodies, improved motor function in mice, and had lower pro-inflammatory cytokine levels.
Oh et al. (2017) [95]	A53T α-synuclein overexpressing cells and mice	Mesenchymal stem cells	1 × 10^6^ cells in 200 µL saline in tail	Eukaryotic elongation factor 1A-2 from MSCs increased neuronal survival by improving axonal transport and monitoring α-synuclein pathological phosphorylation
Kojima et al. (2018) [96]	6-OHDA mice	Catalase mRNA in designer exosomes	Four hundred microliters of the cell/Matrigel mixture	Reduced neuroinflammation and neurotoxicity in Parkinson’s mice.
L’Episcopo et al. (2018) [97]	MPTP mice	Neural stem cells	100 × 10^3^ cells	NSCs that became astrocytic expressed Wnt1 and prompted Wnt/ β-catenin signaling in substantia nigra pars compacta midbrain dopaminergic neurons and microglia. This allowed for dopaminergic neuron rescue and decreased microglial inflammation.
Lee et al. (2019a) [98]	MPTP mice	Human umbilical cord blood stem cells	500 µL cord blood plasma	Mice showed improved motor and GI function, ameliorated dopamine cell loss, and reduced neurological and GI inflammation.
Lee et al. (2019b) [99]	6-OHDA mice	Human umbilical cord blood stem cells	Three separate doses of 4 × 10^6^ cells	Mice showed improved motor and GI function, ameliorated dopamine cell loss, and reduced neurological and GI inflammation.
Serapide et al. (2020) [100]	MPTP Mice	Engrafted astrocytes	150 × 10^3^ ventral midbrain-Astrocytes	Grafted astrocytes can rescue dying dopaminergic neurons, likely via antioxidant and anti-inflammatory Nrf2/ARE/Wnt/β-catenin signaling.

## Data Availability

Not applicable.

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
