# Peer review of "Sequestration of Inflammation in Parkinson’s Disease via Stem Cell Therapy"

_ijms, 2022, doi:10.3390/ijms231710138_

Round 1

Reviewer 1 Report

This is an outstanding review article which comprehensively documents neuroinflammation as a key pathology in PD. The English is excellent and the references are appropriate. This paper is of great value to both preclinical and clinical investigators as neuroinflammation has been overlooked as an important piece of the PD puzzle. Even though I recommend acceptance in the present form, I believe it would be of value to add a small section regarding neurotrophic factors, particularly GDNF. The factor shows neuroprotective/neurorestorative properties in preclinical models and is upregulated by exercise as shown by Zigmond et al. GDNF may also facilitate stem cell survival and function, in line with the authors’ central hypothesis.

Author Response

Dear Editors:

Thank you so much for your constructive comments and allowing for the opportunity to improve our work. We have amended our manuscript to reflect the changes and concerns of the reviewers. Below is our point-by-point response to the review, with specific critiques and concerns directly quoted in italics, followed by our response.

“I believe it would be of value to add a small section regarding neurotrophic factors, particularly GDNF. The factor shows neuroprotective/neurorestorative properties in preclinical models and is upregulated by exercise as shown by Zigmond et al. GDNF may also facilitate stem cell survival and function”

Response: Thank you for your suggestion. A small section, entitled “Neurotrophic factors and stem cell therapy,” has been added to the manuscript (section 3.4). Additionally, the application of exercise and GDNF for endogenous stem cell survival and function (Parkinsonism Relat Disord. 2009 Dec; 15(Suppl 3):S42-5) has been included in section 4.1.

Reviewer 2 Report

In this paper, the authors discuss the role of inflammation in Parkinson’s disease (PD), current therapies, cell-based therapies, and the influence of lifestyle factors on neuroinflammation and cell-based therapies. This review provides direction for future study.

However, the quality of the paper can be improved by add more information related to cell-based therapies because the title of this paper is “Sequestration of Inflammation in Parkinson’s Disease Via Stem Cell Therapy”.

Other comments:

  1. Stem cells are not labeled in Figure 1. The roles of stem cells are not clearly shown in this Figure.
  2. In Section 3.2, first paragraph and second paragraph are not well connected. the “pre-clinical anti-inflammatory stem cell” come out abruptly.  Please expend this paragraph to describe some details of anti-inflammatory stem cell therapies. It will help making transaction to next section as well.
  3. For Section 3.3., it is not clear what is this section aimed to discuss. The authors should reorganize this section either focused on cell-types or pathways.
  4. The title of Section 4 is confusing. What is the connection of this Section with stem cells therapy?
  5. In Sections 4.1 and 4.2, please discuss their effects on neuroinflammation, endogenous stem cells and exogenous stem cell separately.

Author Response

Dear Editors:

Thank you so much for your constructive comments and allowing for the opportunity to improve our work. We have amended our manuscript to reflect the changes and concerns of the reviewers. Below is our point-by-point response to the review, with specific critiques and concerns directly quoted in italics, followed by our response.

“1. Stem cells are not labeled in Figure 1. The roles of stem cells are not clearly shown in this Figure.”

Response: Thank you for your suggestion. Figure 1 had been amended to clearly label stem cells in order to clearly show their role in therapy for Parkinson’s disease.

“2. In Section 3.2, first paragraph and second paragraph are not well connected. the “pre-clinical anti-inflammatory stem cell” come out abruptly.  Please expend this paragraph to describe some details of anti-inflammatory stem cell therapies. It will help making transaction to next section as well.”

Response: We agree that this section was disconnected, we have ameliorated this disconnect by rearranging and supplementing the sections as follows:

“3.2. Cell-based therapies

Another promising route for inducing anti-inflammatory responses in PD may use stem cells. Many of the therapeutic effects of stem cells arise from their inherent secreted anti-inflammatory signals among other cell survival molecules, deemed the bystander effects [91]. Mesenchymal stem cells (MSCs), for example, demonstrate anti-inflammatory, anti-apoptotic, and regenerative capabilities [76,92]. ESC-MSCs and iPSC-MSCs can inhibit lymphocyte and natural killer cell proliferation, also showing promise for immunoregulatory cell-based therapies [93–96]. As noted above, exosomes show promise for PD therapies [76]. MSC-derived exosomes have anti-inflammatory, antioxidant, and neurotrophic properties, ultimately leading to functional recovery in animal models of PD [97,98]. An appealing feature of stem cell treatments includes their ability to be modified prior to implantation, for instance, to promote anti-inflammatory actions. A pathway of interest includes interferon-g signaling via the aforementioned gene, LRRK2. LRRK2 G2019S is the most common genetic mutation associated with inherited PD. In PD patient derived iPSCs possessing LRRK2 G2019S, IFN-g signaling increases immunomodulated neuronal damage [99]. MSCs are initially activated by IFN-g but then exert anti-inflammatory properties [100]. Thus, MSCs may effectively modulate this pathological signaling.

Furthermore, recognizing the underlying DA neuron loss in PD, it is logical to develop treatments which may reestablish these neuronal populations and connections. Cell-based therapies, however, show mixed results in PD, suggesting additional an-ti-inflammatory enhancements of the cells may improve their viability. The use of grafts to restore DA neurons shows mixed results, with some reports demonstrating survival, innervation, and improved functionality, and others showing no clinical benefit [65–73]. The studies demonstrating therapeutic value of stem cells for PD, however, prompt on-going investigation [74,75]. For instance, exosomes from human umbilical cord mesenchymal stem cells (hucMSCs) reduce apoptosis and encourage proliferation in an in vitro PD cell culture. Translated to rats, hucMSC treatment shows similar prevention of DA neuron loss and recovered DA levels [76]. To modulate the altered DA signaling, GABAergic cells from the medial ganglionic eminence (MGE) can be targeted as well. MGE cells show potential to migrate and differentiate into GABAergic neurons, restoring function to mice models treated with 6-hydroxydopamine (6-OHDA), a selective neurotoxin of DA neurons to mimic PD [77]. Human embryonic stem cells (hESCs) in rat models of PD can project long distances in the rodent brain to mitigate symptoms, offering hopeful clinical translation [78]. Even closer to the goal of clinical application, human parthenogenetic embryonic stem cell (hPESC) derived DA neurons improve mobility in monkey models of PD [79]. Also, in monkey 1-methyl-4-phenyl-1,2,3,6-tetrahydropyridine (MPTP) models recreating PD, human induced pluripotent stem cells (iPSCs) improve movement and demonstrate ample neurite formation. Whether grafted from PD or control donors, iPSCs survive, make connections, and lead to functional recovery in monkeys [80]. There is still room for improvement for stem cell PD treatments, as full recovery is not typically achieved in animal models [81–83].”

“3. For Section 3.3., it is not clear what is this section aimed to discuss. The authors should reorganize this section either focused on cell-types or pathways.”

Response: In addressing your prior suggestion, we hope to have made this section more focused as well. We moved the original discussion on cell types to 3.2 and added a stronger topic sentence to set this section’s focus on cell signaling pathways aimed at enhancing the cells’ immunomodulatory properties. The section now begins as follows:

“While already anti-inflammatory by nature, stem cell treatments may be further enhanced in clinical effectivity and viability by modulating these cells to express additional anti-inflammation properties. The potential for MSCs to impact cell signaling can also be applied to the Wnt/β-catenin pathway.”

“4. The title of Section 4 is confusing. What is the connection of this Section with stem cells therapy?”

Response: Thank you for your suggestion. The title of section 4 has been changed to “Lifestyle factors facilitating the stem cell response” to clarify the connection between exercise/diet and stem cell functions such as sequestration of neuroinflammation, neuroprotection, and neurogenesis.

“5. In Sections 4.1 and 4.2, please discuss their effects on neuroinflammation, endogenous stem cells and exogenous stem cell separately.”

Response: Thank you for your suggestion. Sections 4.1 and 4.2 have been reorganized so that the effects of exercise/diet on neuroinflammation, endogenous stem cells, and exogenous stem cells are presented separately.